# Effect of Anti-Programmed Cell Death-1 Antibody on Middle Ear Mucosal Immune Response to Intranasal Administration of *Haemophilus influenzae* Outer Membrane Protein

**DOI:** 10.3390/vaccines13030313

**Published:** 2025-03-13

**Authors:** Kazuhiro Yoshinaga, Takashi Hirano, Shingo Umemoto, Yoshinori Kadowaki, Takayuki Matsunaga, Masashi Suzuki

**Affiliations:** Department of Otorhinolaryngology& Head and Neck Surgery, Faculty of Medicine, Oita University, Oita 879-5593, Japan; k-yoshinaga@oita-u.ac.jp (K.Y.);

**Keywords:** acute otitis media, PD-1, PD-L1, humoral immunity, T follicular helper cells

## Abstract

**Background/Objectives**: Acute otitis media is a common pediatric infection caused primarily by nontypeable *Haemophilus influenzae*. With rising antibiotic resistance, vaccines are essential for combating this public health issue. Although the PD-1/PD-L1 pathway has been extensively studied for its role in tumor immunity, its impact on mucosal immunity, particularly in vaccine responses, is unclear. **Methods**: BALB/c mice were intranasally immunized with nontypeable *H. influenzae* outer membrane protein and treated with anti-PD-L1 antibodies. Immune responses were evaluated in middle ear mucosa (MEM), the cervical lymph node, and the spleen using an enzyme-linked immunosorbent assay, an enzyme-linked immunospot assay, and flow cytometry. The effects on CD4^+^ T cells, T follicular helper (Tfh) cells, and B-cell differentiation were analyzed. **Results**: Anti-PD-L1 antibody treatment increased CD3^+^CD4^+^CD185^+^ (CXCR5^+^) Tfh cells in MEM, which play a crucial role in supporting B-cell activation and antibody production. This correlated with a significant increase in IgA- and IgG-producing cells in MEM, which enhanced local bacterial clearance. Although B-cell activation and differentiation into plasmablasts were observed in MEM, no significant changes were noted in the cervical lymph node and spleen, suggesting a localized enhancement of mucosal immunity. **Conclusions**: Anti-PD-L1 antibodies promoted Tfh cell expansion and B-cell differentiation in MEM, leading to enhanced antibody production and improved bacterial clearance. These findings suggest that PD-L1 blockade can potentiate mucosal vaccine-induced immunity by strengthening local humoral responses. This supports its potential application in developing intranasal vaccines for acute otitis media.

## 1. Introduction

In the US, by the age of 1, about 20% of children will have experienced one or more episodes of AOM, 60% of children will have at least one acute otitis media infection by age 3 years, and 80% will have at least one acute otitis media infection in their lifetime [1,2]. Thus, AOM is one of the most common infectious diseases in children. Its prevalence peaks in early childhood and is associated with consequences such as hearing loss, delayed language development, permanent damage to the middle ear, and mucosal changes. Nontypeable *Haemophilus influenzae* (NTHi) is a bacterial pathogen and a major cause of AOM and respiratory infections [3]. Current treatment is based on the use of antibiotics, which face serious challenges due to the emergence of antibiotic-resistant bacteria [4,5]. Therefore, the development of vaccines is an urgent public health goal.

Programmed cell death-1 (PD-1: CD279) is an immunoreceptor identified in 1992. It is an immunosuppressive cosignaling receptor that belongs to the immunoglobulin superfamily, based on molecular structure [6]. PD-1 is expressed in activated T cells, B cells, and myeloid cells and suppresses T-cell proliferation and effector function [7]. Programmed cell death-1 ligand 1 (PD-L1), a ligand of PD-1, is constitutively expressed in macrophages and vascular endothelial cells, and its expression is increased by stimuli, such as inflammation [8]. Tumor cells and virus-infected cells also express PD-L1, and the proliferation and damage function of T cells is suppressed by PD-1/PD-L1 signaling, allowing them to escape immune surveillance.

In recent years, antibody therapy targeting immune checkpoint molecules has been widely used as an immunotherapy to activate tumor immune responses in various malignancies. For example, PD-L1 inhibitors effectively treat advanced solid tumors, including melanoma, nonsmall cell lung cancer, renal cell carcinoma, castration-resistant prostate cancer, and colorectal cancer [9]. Since 2016, PD-L1 inhibitors have been used to treat head and neck cancer [10]. Administration of antibodies that bind PD-1 on T cells or PD-L1 (expressed by malignant tumor cells or antigen-presenting cells) inhibits interaction with PD-1 on T cells, thereby inhibiting inhibitory signaling to T cells, maintaining the activation of CD8^+^ cytotoxic T cells, and exerting antiviral and antitumor effects [11]. Thus, reactivation of immune responses mainly in CD8^+^ T cells by PD-1/PD-L1 signaling inhibition has been reported. While most studies have focused on the role of PD-L1 inhibition in enhancing tumor immunity, new evidence indicates that the PD-1/PD-L1 blockade can also modulate vaccine responses. For example, a study on patients with cancer receiving immune checkpoint inhibitors showed that influenza vaccination increased viral antibody titers, when compared with that in healthy individuals, with a substantially higher seroconversion index [12]. This suggests that PD-L1 inhibition may enhance vaccine-induced immune responses by promoting antibody production. However, the effects of inhibiting PD-1/PD-L1 signaling on mucosal immunity, particularly concerning IgA-mediated responses crucial for mucosal vaccine efficacy, remain largely unexplored. Herein, we investigate the effects of PD-1/PD-L1 signaling inhibition via anti-PD-L1 antibody administration on CD4^+^ T cells and B cells in tissues from a mouse vaccine model of otitis media, focusing on its potential role in enhancing mucosal immune responses.

## 2. Materials and Methods

### 2.1. Animals

BALB/c mice were obtained from Kyudo Japan (Fukuoka, Japan). Mice were maintained in a pathogen-free facility until 6 weeks of age, at which time they were used in the experiments. A total of 150 mice were used in this study. All experiments were approved by the Animal Care and Use Committee of Oita University (No. 232801, and No. 182802).

### 2.2. Immunogen and Adjuvant

NTHi (strain 76), originally isolated from the nasopharynx of a patient with otitis media with effusion at Oita University, was stored at −80 °C and used for antigen and nasal inoculum preparation. The antigen was obtained by a previously described method [13]. Briefly, NTHi was grown on chocolate agar at 37 °C under 5% CO_2_ overnight. Bacterial colonies were collected from the plate with a sterile loop, resuspended in EDTA buffer (pH 7.4), and subjected to heat treatment at 56 °C for 30 min. The bacterial suspension was then sonicated on ice to facilitate cell disruption. Intact cells and major debris were then removed by centrifugation at 10,000× *g* for 20 min. The resulting supernatant was further processed by ultracentrifugation at 80,000× *g* for 2 h at 4 °C. The final pellet, which appeared as a clear gel-like substance, was resuspended in distilled water and subjected to lyophilization. The resulting powder, referred to as the outer membrane protein (OMP) fraction of NTHi, was stored at 4 °C until required for use [14].

### 2.3. Intranasal Immunization Mouse Model

In total, 10 μg OMP from NTHi strain 76 and 1 μg cholera toxin (as mucosal adjuvant) were dissolved in phosphate-buffered saline (PBS) to prepare OMP solution (10 μL). OMP was prepared according to the method of Murphy et al. [14]. Vaccine quality control was performed using OMP per previously described protocols published by our department [13,15,16]. Briefly, the protein concentration of each batch was first determined using a bicinchoninic acid assay to ensure consistency. In addition, SDS-PAGE was conducted to confirm the purity and integrity of OMP. Cultured NTHi were incubated in EDTA solution at 56 °C for 30 min. The cell membranes were disrupted by ultrasound on ice. The lysed cells were centrifuged to extract the OMP solution. The OMP dose (10 μg) was determined based on previous studies of NTHi OMP immunization in murine models, particularly those of Hirano et al. [13] and Iwasaki et al. [15], demonstrating that this concentration conferred optimal immunogenicity and safety. Six-week-old mice (*n* = 5) were intranasally immunized with OMP solution once a week three times. On day 7 after the end of intranasal immunization, the mice were deeply anesthetized with ketamine–xylazine solution. Blood was collected and cervical dislocation was performed. Tympanotomy was performed under a microscope, and the inside of the tympanic cavity was washed with 200 μL of physiological saline to collect middle ear washings (MEWs) (Figure 1).

### 2.4. PD-L1 Antibody Delivery Model

Channappanavar et al. set the dose of anti-PD-L1 antibody (10F.9G2) at 300 μg and administered it intraperitoneally on day 1 before and day 3 after acute herpes simplex virus infection and reported an increase in HSV-specific effector CD8^+^ T-cell responses [17]. In this study, we followed the method of Channappanavar et al. for the dose and administration of anti-PD-L1 antibody and administered 250 μg of anti-PD-L1 antibody (10F.9G2: GeneTex, CA, USA) intraperitoneally on day 1 before and day 3 after immunization. We immunized each mouse (*n* = 5) with OMP solution once a week three times intranasally. Samples were collected on day 7 after nasal immunization, as described (Figure 1).

### 2.5. Detection of OMP-Specific Antibodies by Enzyme-Linked Immunosorbent Assay

Specific anti-OMP antibody levels in MEWs and serum samples were quantified by enzyme-linked immunosorbent assay (ELISA) using OMP as the coating antigen at a concentration of 10 µg/mL. Samples collected from naive mice on day 0 served as negative controls and resulted in optical density (A405) values of less than 0.1 for IgA and IgG in serum and less than 0.01 in body fluids. The endpoint titer for anti-OMP antibody was established as the highest sample dilution that resulted in an optical density at least twice that of the negative control. The determination of anti-OMP antibody levels in MEWs and serum followed this criterion, ensuring that the absorbance was greater than twice that of the naive mouse sample, as described by Jiao et al. [18].

### 2.6. Flow Cytometry Analysis

To analyze the cellular phenotypes of lymphocytes in the middle ear mucosa (MEM), cervical lymph nodes (CLN), and spleen (SPL) following OMP immunization, with and without anti-PD-L1 antibody administration, we isolated mononuclear cells (MNCs) and stained them with antigen-specific antibodies conjugated to fluorescent labels. Isolation of MNCs from MEM was performed using a previously established tissue dissociation protocol [19]. Briefly, the skin covering the head was carefully removed and the bulla was dissected from the temporal bone. Under a microscope, the MEM was separated from the bone structure. The extracted tissue was then enzymatically digested with 0.5 mg/mL collagenase type IV (Sigma, St. Louis, MO, USA) at 37 °C for 10 min to release MNCs. Approximately 0.5–1.0 × 10^6^ cells were used for flow cytometry. To detect T cells and B cells in MEM, CLN, and SPL, FITC-conjugated anti-CD3 monoclonal antibody (mAb) (145-2C11; BD Biosciences, San Jose, CA, USA), Pacific Blue-conjugated anti-CD4 mAb (GK1.5; BD Biosciences), APC/cyanine7 -conjugated anti-CD8 mAb (53-6.7; BD Biosciences), PE-conjugated anti-CD44 mAb (IM7; BD Biosciences), PE/cyanine7-conjugated anti-CD62L mAb (MEL-14; BD Biosciences), PE/cyanine7-conjugated anti-CD19 mAb (6D5; BD Biosciences), APC-conjugated anti-CD22 mAb (OX-97; BD Biosciences), FITC-conjugated anti-B220 mAb (RA3-6B2; BD Biosciences), and PE-conjugated anti-CD185+ mAb (L138D7; BD Biosciences) were used to analyze T- and B-cell subsets. The corresponding isotype control antibodies were used as negative controls for background staining. The immunofluorescence intensity of T and B cells was analyzed using LSR Fortessa X-20 (BD Biosciences, San Jose, CA, USA). Cells were stained with fluorescently-labeled antibodies specific for T- and B-cell markers: anti-CD3 for T cells, anti-CD4 and -CD8 for T-cell subsets, and anti-CD19 and B220 for B cells. Next, we used anti-CD22 and -CD185 antibodies to assess B-cell differentiation. After staining, flow cytometry was performed using an LSR Fortessa X-20 (BD Biosciences). Lymphocytes were first gated based on forward and side scatter properties. Next, B cells within this population were identified via examination of CD19 and B220 expression, and their differentiation status was analyzed using CD22 and CD185 markers. All acquired data were processed using FlowJo (Tree Star, Ashland, OR, USA).

### 2.7. Enzyme-Linked Immunospot Assay

The measurement of OMP-specific IgA- and IgG-producing cells in the MEM was performed using enzyme-linked immunospot (ELISPOT) assay. To capture OMP antigen-specific antibodies, OMP was coated on the bottom of 96-well filtration plates with a nitrocellulose base (Millititer HA, Millipore Corp., Bedford, MA, USA). Mouse MEM cells from the OMP- and PD-L1-treated groups were attached to the coated plate and cultured for 24 h. This resulted in the secretion of antibodies, such as IgA and IgG, which bind to the antigen. The captured IgA and IgG were detected using biotinylated anti-mouse IgA and IgG antibodies, respectively. Spots were visualized by HRP enzyme labeling and substrate reaction, and the number of OMP-specific IgA-producing cells was quantified by counting under a light microscope.

### 2.8. Bacterial Challenge and Sampling

A separate cohort of three groups of mice was established for the bacterial challenge assay. NTHi strain 76 was used as the challenge organism. Bacteria were cultured on chocolate agar at 37 °C with 5% CO_2_ for 16 h. The bacterial density was determined by assessing optical absorbance at 600 nm. A bacterial suspension was prepared at a concentration of 1.0 × 10^6^ colony-forming units (cfu)/mL in PBS and kept on ice until use. Each mouse received an inoculation of 10 µL of the prepared NTHi suspension (10^4^ cfu/mouse) directly into the middle ear bulla. One day after challenge, 5 to 10 mice per group were anesthetized with ketamine and xylazine, and blood was collected from the axillary artery. After decapitation, MEW samples were collected via myringotomy according to the previously described procedure. The bacterial load in the MEWs was determined using established microbiological quantification methods [15].

### 2.9. Statistical Analysis

Statistical analyses for comparisons among multiple groups were conducted using the Kruskal–Wallis test, considering a *p*-value of <0.05 as the threshold for significance. When significant differences were detected, pairwise comparisons were conducted using the Steel–Dwass test, with a significance level set at *p* < 0.05. To evaluate differences between groups with and without anti-PD-L1 antibody administration, the Mann–Whitney test was applied, with a *p*-value of <0.05 regarded as statistically significant.

## 3. Results

### 3.1. Changes in OMP-Specific Antibody Titers and Effects of Anti-PD-L1 Antibody Administration

The level of OMP-specific antibodies in MEWs and serum was measured by ELISA.

Compared to the control group, the IgA levels in the MEWs were significantly higher in the OMP and anti-PD-L1 antibody groups (4.6 ± 2.0 vs. 0 ± 0 and 5.2 ± 3.5 vs. 0 ± 0: log2 titer, respectively; *p* < 0.01) (Figure 2a). Although the difference between the OMP and anti-PD-L1 antibody groups was insignificant, the antibody titer was higher in the anti-PD-L1 antibody group. However, the differences in the serum IgA levels between the two groups were not significant.

IgG levels in the MEWs were significantly higher in the OMP and anti-PD-L1 antibody groups compared with the control group (2.3 ± 1.4 vs. 0 ± 0 and 3.5 ± 1.3 vs. 0 ± 0: log2 titer, respectively; *p* < 0.01) (Figure 2a). Serum IgG levels were significantly higher in the OMP and anti-PD-L1 antibody groups than in the control group (9.7 ± 2.1 vs. 0 ± 0, 12.5 ± 0.87 vs. 0 ± 0: log2 titer, respectively; *p* < 0.01) (Figure 2b) and significantly higher in the anti-PD-L1 antibody group than in the OMP group (9.7 ± 2.1 vs. 12.5 ± 0.87: log2 titer, respectively; *p* < 0.01) (Figure 2b).

### 3.2. Flow Cytometry Analysis

#### 3.2.1. Effect of Anti-PD-L1 Antibody Administration on the CD3+ T-Cell Fraction

Total MNC counts of MEM, CLN, and SPL were measured in the control, OMP(+), and PD-L1(+) groups, respectively, and no significant differences in total MNC counts were observed in any case (Table 1). To investigate the effect of anti-PD-L1 antibody administration after nasal immunization with OMP, we gated the lymphocyte region of MNCs collected from MEM, CLN, and SPL and analyzed the CD3^+^ T-cell fraction. In MEM, no change in CD3^+^ cells was observed after the administration of anti-PD-L1 antibody. In CLN and SPL, the anti-PD-L1 antibody group tended to have a lower proportion of CD3^+^ cells than the control group, although the difference was insignificant. Notably, in CLN, considerable variability was observed within the control group, leading to a higher mean value. However, no statistically considerable differences were detected between the control and PD-L1 groups (Figure 3).

We examined the expression of CD4 and CD8 in CD3^+^ cells. In MEM, CD4^+^CD8^−^ cells tended to be higher with OMP treatment alone, but the difference was insignificant. No change was observed in the anti-PD-L1 antibody group compared with the control group.

In the anti-PD-L1 antibody group, differences in the number of CD4 and CD8^+^ cells were insignificant. Although there was no significant difference in CD4^−^CD8^−^ cells, there was a tendency for them to be lower after anti-PD-L1 antibody administration compared with the control group.

In CLN, the proportion of CD4−CD8+ cells was significantly higher in the anti-PD-L1 antibody group than in the control group (0.66 ± 0.38 vs. 1.48 ± 0.36; *p* < 0.05) (Figure 4b).

#### 3.2.2. Analysis of CD4^+^ T-Cell Dynamics After Administration of Anti-PD-L1 Antibody

CD3^+^CD4^+^ cells were further gated for CD44 and CD62L, and memory T cells, effector T cells, and naive T cells were examined.

Memory T cells in MEM were significantly more abundant in the OMP group than in the control group. The difference was insignificant in the anti-PD-L1 antibody group, but the rate tended to be higher. Effector T cells tended to be lower in the OMP and anti-PD-L1 groups than in the control group, but the difference was insignificant. Naive T cells tended to be higher in the OMP and anti-PD-L1 antibody groups than in the control group, but the difference was insignificant (Figure 5a).

In CLN, the anti-PD-L1 antibody group showed a significantly higher rate of memory T cells than the control group (68.04 ± 7.9 vs. 10.4 ± 9.7; *p* < 0.01) (Figure 5b), the anti-PD-L1 antibody group showed a significantly lower rate than the OMP group in effector T cells, and the anti-PD-L1 antibody group showed a significantly higher rate than the control group in naive T cells (14.1 ± 3.7 vs. 3.1 ± 2.4; *p* < 0.01) (Figure 5b). In SPL, the anti-PD-L1 antibody group showed a significantly higher rate than the control group in memory T cells (29.3 ± 4.5 vs. 12.7 ± 5.7; *p* < 0.01) (Figure 5c), whereas the anti-PD-L1 antibody group showed a significantly lower rate than the control and OMP groups in effector T cells (7.8 ± 4.0 vs. 28.6 ± 10.4, 7.8 ± 4.0 vs. 38.6 ± 20.9; *p* < 0.01) (Figure 5c). In naive T cells, there was no difference between the three groups.

#### 3.2.3. Analysis of B-Cell Dynamics Following Anti-PD-L1 Antibody Administration

Anti-PD-L1 administration resulted in significantly higher CD19+ cells in MEM and SPL than in control (32.2 ± 4.5 vs. 48.46 ± 8.8, 19.52 ± 2.7 vs. 32.8 ± 7.0, respectively; *p* < 0.05) (Figure 6). In CLN, the proportion of CD19+ cells did not change between the control and anti-PD-L1 antibody groups.

CD19^+^ cells were gated using CD22 and B220. The differentiation of B cells, such as plasmablast cells, mature B cells, and immature B cells, was analyzed.

The rate of plasmablast cells was significantly higher in the OMP group than in the control group (87.7 ± 7.1 vs. 68.0 ± 12.3; *p* < 0.05) (Figure 7a). In the anti-PD-L1 antibody group, the rate tended to be higher than that in the control group, although the difference was insignificant. In mature cells, the difference was insignificant compared with the control group. In immature cells, the rate was significantly lower in the anti-PD-L1 antibody group than in the control group (3.1 ± 4.1 vs. 6.2 ± 3.5; *p* < 0.05) (Figure 7a).

In CLN, no significant differences were observed between the three groups in terms of plasmablast cells, mature B cells, or immature B cells.

In SPL, although plasmablast cells tended to be lower in the anti-PD-L1 antibody group than in the control group, the difference was not significant. The rate of mature B cells was significantly higher in the anti-PD-L1 antibody group than in the control and OMP groups (3.1 ± 4.1 vs. 6.2 ± 3.5; *p* < 0.05) (Figure 7c).

#### 3.2.4. Analysis of the Dynamics of CD3^+^CD4^+^CD185^+^ T Cells After the Administration of the Anti-PD-L1 Antibody

CD185^+^ cells were examined in CD3^+^CD4^+^ cells in MEM. The percentages were low in the control and OMP groups, with no significant difference. However, the percentage of CD3^+^CD4^+^CD185^+^ T cells was significantly higher in the anti-PD-L1 antibody group than in the control and OMP groups (0.58 ± 1.3 vs. 5.0 ± 3.6, 0.32 ± 3.7vs. 5.0 ± 3.6; *p* < 0.05) (Figure 8).

### 3.3. Measurement of Antibody-Producing Cells by ELISPOT Asssay with Administration of Anti-PD-L1 Antibody

ELISPOT quantification of OMP antigen-specific IgA and IgG antibody-producing cells in the MEM by anti-PD-L1 antibody administration. Compared to the control group, the number of IgA and IgG antibody-producing cells was higher in the OMP and anti-PD-L1 antibody groups. (Figure 9).

### 3.4. Bacterial Clearance

The number of bacteria counted after culturing the MEWs tended to be lower in the OMP group than in the control group, although the difference was insignificant. However, the number of bacteria was significantly lower in the anti-PD-L1 antibody group, indicating increased bacterial clearance (Figure 10).

## 4. Discussion

PD-1 is an inhibitory receptor induced on T cells by antigen stimulation. The persistent expression of PD-1 suppresses T-cell function. Blocking PD-1 signaling is an effective cancer immunotherapy for malignant tumors because it reactivates exhausted T cells [20,21]. Blocking the PD-1/PD-L1 pathway has been reported as a potential treatment for chronic infections. Barber et al. reported that the administration of anti-PD-L1 antibodies to mice chronically infected with lymphocytic choriomeningitis virus increased the number of lymphocytic choriomeningitis virus-specific CD8^+^ T cells, restored cytokine production and cytotoxicity, and promoted viral control [19]. Velu et al. conducted a study in macaques chronically infected with a similar immunodeficiency virus and reported that PD-1 blockade increased memory B-cell proliferation and SIV envelope-specific antibodies, enhancing cellular and humoral immune responses to SIV [22]. Reportedly, a PD-1 blockade can modulate humoral immunity under chronic infection conditions. However, its role in vaccine-induced immunity, particularly during the early T-cell response to primary immunization remains unexplored. Although most studies have focused on the CD8^+^ T-cell response, the effects on B-cell differentiation and CD4^+^ T-cell immunity concerning vaccination have received less attention.

For example, Läubli et al. reported that immune checkpoint inhibitors enhanced humoral immune responses in patients with cancer receiving influenza vaccines [12]. Consistent with those findings, this study demonstrated that intranasal vaccination combined with PD-L1 blockade resulted in a strong humoral immune response. This suggests that PD-L1 inhibition can be a promising strategy for enhancing the efficacy of human intranasal vaccines. However, further studies, including human clinical trials, are required to validate these results.

Nishimura et al. generated PD-1-deficient mice and reported an increase in serum IgG2b and IgA in PD-1-deficient mice and increase in specific antibody production response of B lymphocytes to T-dependent antigens [23]. This is thought to support the effect of blocking the PD-1 pathway in nasal immunization of 6-week-old mice, which enhances OMP-specific antibody production.

Consistently, ELISA showed that IgA and IgG in the MEWs were significantly higher than in the control group, suggesting that PD-L1 administration enhanced the production of IgA and IgG antibodies in MEM, which is a local tissue. Although serum IgA level did not change, serum IgG level was significantly higher, suggesting that PD-L1 antibody administration increased the production of IgG antibodies in the whole body.

Therefore, we performed phenotypic analysis and functional evaluation of the immune cells by flow cytometry, and the percentage of CD3^+^ cells in the MEM, CLN, and SPL did not change after administration of anti-PD-L1 antibody compared to the control group, but in the CLN, the percentage was significantly lower than in the OMP-treated group. It has been reported that administration of anti-PD-L1 antibodies increases the percentage and absolute number of CD3+ T cells in the tumor microenvironment in young animal models [24]. In this study, the percentage and number of CD3^+^ cells did not change in any tissue after PD-L1 administration compared to the control, suggesting that the response may be tissue dependent.

On the other hand, among memory T cells, effector T cells and naive T cells among CD4+ cells, memory T cells and naive T cells were significantly higher in the CLN than in the control group after PD-L1 administration, and memory T cells and naive T cells tended to be higher in MEN and SPL. The percentage of effector T cells was significantly reduced in SPL, and although the difference was insignificant, we also observed a tendency for this percentage to decrease in MEN and CLN. The generation of tissue-resident memory T cells is an essential component of the immune response on mucosal surfaces. Reportedly, the preferential generation of tissue-resident memory T cells is a principal advantage of mucosally administered vaccines [25].

Fanelli et al. suggest that blocking the PD-1/PD-L1 pathway counteracted this inhibitory signaling and promoted CD4^+^ effector or memory T-cell proliferation [26].

These changes suggest that antigen stimulation by multiple administrations of PD-L1 + OMP led to the accumulation of several memory T cells, which, in turn, led to a relative decrease in the proportion of effector T cells.

The anti-PD-L1 antibody resulted in predominantly more CD19^+^ cells in the MEM, with a trend toward a higher proportion of plasmablasts, although the difference was insignificant. The proportion of immature B cells in MEM decreased, suggesting that activated B cells differentiate and convert from immature B cells in MEM and plasmablasts and plasma cells tend to increase, i.e., antibody production is enhanced. By contrast, in lymphoid tissues, such as CLN and SPL, the number of mature B cells tended to increase and the proportion of immature cells decreased, although studies are lacking on the dynamics of local and systemic B-cell activation by PD-L1 administration, suggesting differences in local and systemic immune responses.

In this study, administration of anti-PD-L1 antibodies increased humoral immunity in the MEM, which is a local immune response, and promoted the systemic humoral immune response. These findings suggest that anti-PD-L1 antibodies are effective in vaccine development.

Follicular helper T (Tfh) cells help B cells produce antibodies, for example, by maintaining the life of B cells, promoting their proliferation, and encouraging their differentiation into plasma cells [27]. Understanding the role of Tfh cells will lead to better vaccines. In this study, we analyzed CD4^+^ T cells positive for CD185 (CXCR5), a representative marker of Tfh cells, by FACS and found that PD-L1 administration significantly increased CD185^+^CD4^+^ T cells in the MEM [28]. Khan et al. reported that increased PD-L1 expression reduced Tfh cells via Breg cells and regulated humoral immunity [29]. In this study, the administration of the anti-PD-L1 antibody blocked the PD1/PDL-1 pathway, which is expected to reduce PD-L1 expression, resulting in an increase in CD185^+^CD4^+^ T cells, a subtype of Tfh cells, suggesting enhanced humoral immunity in the middle ear.

OMP-specific antigen-producing cells were analyzed using the ELISPOT assay. The marked increase in cells producing IgA and IgG in the MEM suggested that the increase in OMP-specific titer in MEWs observed using ELISA were due to an increase in OMP-specific antigen-producing cells. Moreover, the increased plasmablast fractions observed in CD19^+^ cells of MEM, CLN, and SPL may be involved. Oh et al. reported that intranasal priming of influenza A virus induced local lung-resident B-cell populations that secreted protective mucosal antiviral IgAs, including IgA+ plasma and plasmablast cells. Such tissue-resident IgA-secreting B cells provide insight into the establishment of protective immunity in the airway [30].

In the bacterial clearance test, the number of H. influenzae in the middle ear effluent 24 h after infection with NTHi was predominantly lower in the anti-PD-L1 antibody group than in the control group, suggesting that anti-PD-L1 antibody treatment enhanced H. influenzae clearance.

Based on the results of ELISA, flow cytometric analysis, ELISPOT assay, and bacterial clearance assay, we hypothesize that administration of anti-PD-L1 antibodies stimulated humoral immunity and increased the antibody production capacity to produce OMP-specific antibodies, resulting in local antibody production in the MEM, which, in turn, increased clearance and reduced bacterial count. Future research will continue to translate these findings to intranasal vaccines and promote the development of efficient vaccines.

## 5. Conclusions

Our findings demonstrate that PD-L1 blockade enhances mucosal immunity by increasing antigen-specific IgA production and bacterial clearance in the middle ear mucosa. This suggests that PD-L1 inhibition could be a promising strategy for enhancing the efficacy of intranasal vaccines, particularly for respiratory and middle ear infections. Further studies are warranted to evaluate its potential application in human vaccine development.

## Figures and Tables

**Figure 1 vaccines-13-00313-f001:**
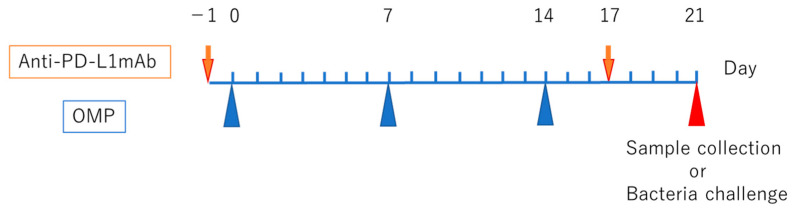
The experimental protocol timeline. Briefly, 6-week-old mice (*n* = 5) were intranasally immunized with an OMP solution once per week for 3 weeks (i.e., on days 0, 7, and 14). Next, an anti-PD-L1 monoclonal antibody (10F.9G2, 250 μg) was administered intraperitoneally 1 day before the first OMP immunization (day −1) and 3 days after final immunization (day 17). On day 21, samples were collected, and a bacterial challenge was performed.

**Figure 2 vaccines-13-00313-f002:**
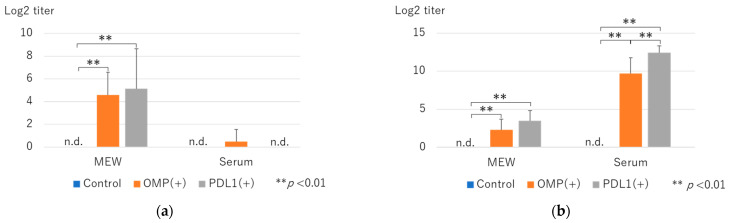
Changes in OMP-specific antibody titers after administration of anti-PD-L1 antibody. (**a**) Log2 titers of OMP-specific IgA antibodies in middle ear washings (MEWs) and serum. MEW antibody titers were significantly higher in the OMP and OMP+ anti-PD-L1 antibody groups relative to the control group (*p* < 0.05). Moreover, IgA was detected in serum in the OMP group; however, we found no significant difference between the control and anti-PD-L1 groups (n.d. indicates not detected). (**b**) Log2 titers of OMP-specific IgG antibody in MEWs and serum. In MEWs and serum, the OMP and anti-PD-L1 antibody groups showed elevated IgG titers relative to the control group (*p* < 0.05). OMP (+): OMP-treated group. PD-L1 (+): OMP+ anti-PD-L1 antibody-treated group. n.d.: not detected. The Mann–Whitney U test was used for statistical analysis. ** *p* < 0.01. Values are presented as mean ± SD.

**Figure 3 vaccines-13-00313-f003:**
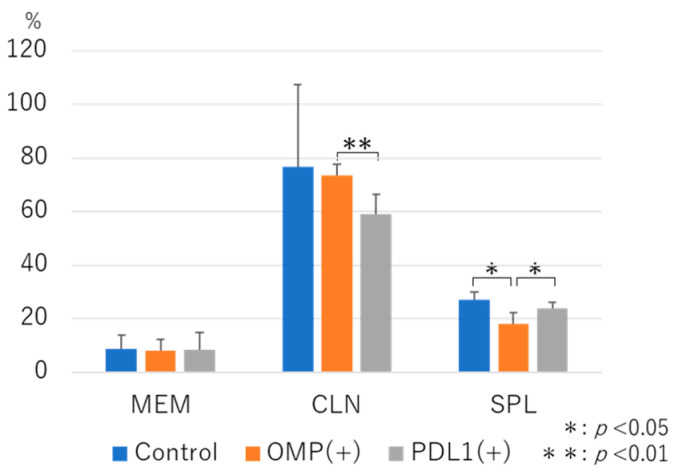
Flow cytometry analysis of total CD3^+^ T cells in various tissues. No significant differences were observed between the anti-PD-L1 antibody group and the control group in middle ear mucosa (MEM), cervical lymph node (CLN), or spleen (SPL) samples. However, in CLN, the anti-PD-L1 antibody group was lower than the OMP group (*p* < 0.01), and in SPL, the OMP group was lower than the control and anti-PD-L1 groups (*p* < 0.01). OMP (+): OMP-treated-group. PD-L1 (+): OMP+ anti-PD-L1 antibody-treated group. The Steel–Dwass test was used for statistical analysis. * *p* < 0.05, ** *p* < 0.01. Values are presented as mean ± SD.

**Figure 4 vaccines-13-00313-f004:**
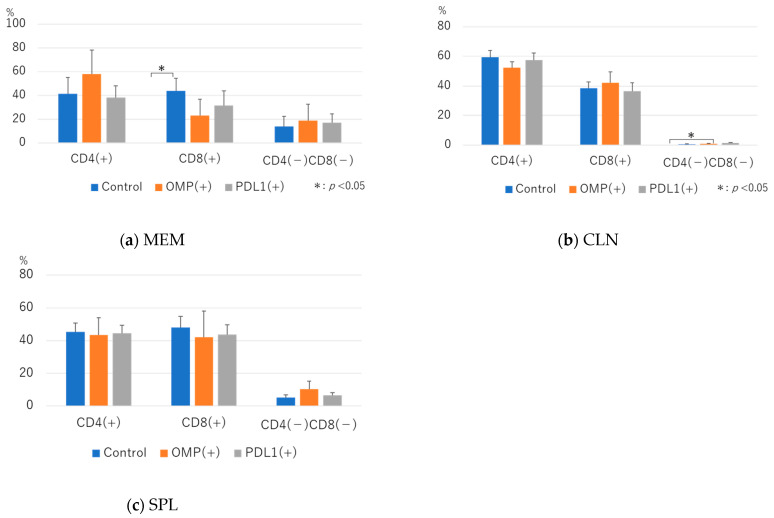
Flow cytometric analysis of CD4^+^ and CD8^+^ T cells in CD3^+^ cells: (**a**) middle ear mucosa (MEM); (**b**) cervical lymph node (CLN); (**c**) spleen (SPL). Flow cytometric analysis of CD4^+^ and CD8^+^ T cells showed no significant differences between the anti-PD-L1 antibody and control groups in MEM, CLN, or SPL. The OMP group had significantly lower CD8^+^ in MEM than the control group, whereas the anti-PD-L1 antibody group had significantly higher CD4-CD8- in CLN, especially in MEM (*p* < 0.01).OMP (+): OMP-treated group. PD-L1 (+): OMP+ anti-PD-L1 antibody-treated group. The Steel–Dwass test was used for statistical analysis. * *p* < 0.05. Values are presented as mean ± SD.

**Figure 5 vaccines-13-00313-f005:**
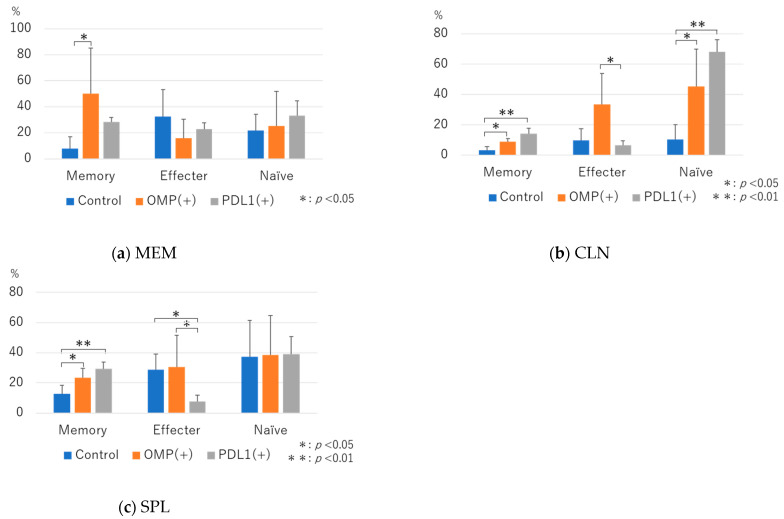
Flow cytometry analysis of CD4^+^ T-cell subsets. CD3^+^CD4^+^ cells were further gated for CD44 and CD62L to evaluate memory T cells, effector T cells, and naive T cells. (**a**) Middle ear mucosa (MEM): memory T cells were significantly increased in the OMP group compared with the control group (*p* < 0.05). (**b**) Cervical lymph node (CLN): memory T cells and naive T cells were significantly more abundant in the anti-PD-L1 antibody group than in the control group (*p* < 0.01). (**c**) Spleen (SPL): memory T cells were significantly more abundant in the anti-PD-L1 antibody group than in the control group (*p* < 0.01), and effector T cells were significantly less abundant (*p* < 0.05). OMP (+): OMP-treated group. PD-L1 (+): OMP+ anti-PD-L1 antibody-treated group. The Steel–Dwass test was used for statistical analysis. * *p* < 0.05, ** *p* < 0.01. Values are presented as mean ± SD.

**Figure 6 vaccines-13-00313-f006:**
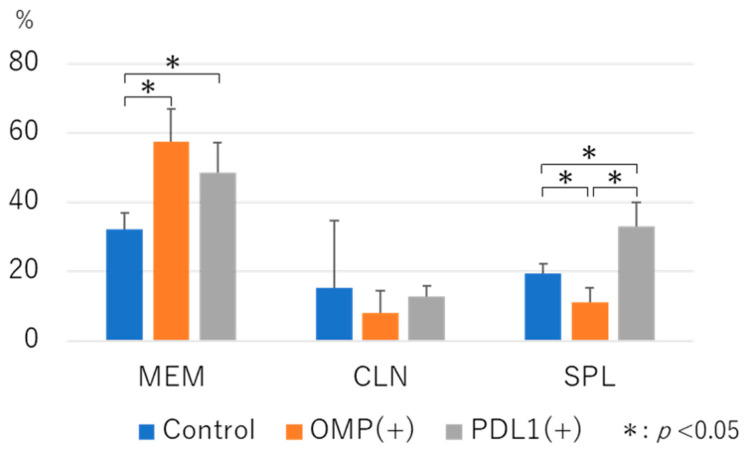
Flow cytometry analysis of CD19^+^ B cells in the tissues: The percentage of CD19^+^ B cells was significantly increased in MEM and SPL in the anti-PD-L1 antibody group compared with the control group (*p* < 0.05). Middle ear mucosa (MEM), cervical lymph node (CLN), spleen (SPL). OMP (+): OMP-treated group. PD-L1 (+): OMP+ anti-PD-L1 antibody-treated group. The Steel–Dwass test was used for statistical analysis. * *p* < 0.05. Values are presented as mean ± 1 SD.

**Figure 7 vaccines-13-00313-f007:**
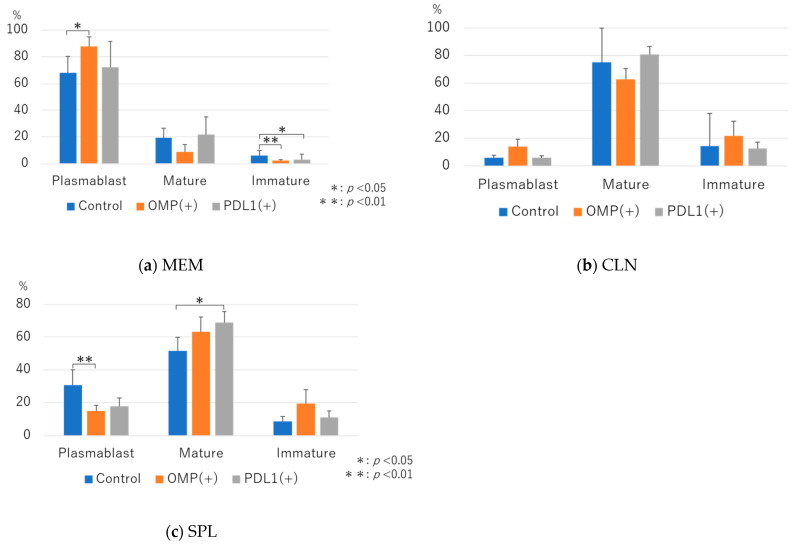
Flow cytometric analysis of B-cell differentiation. CD19^+^ cells were gated using CD22 and B220, and plasmablast cells, mature B cells, and immature B cells were evaluated. (**a**) In the middle ear mucosa (MEM), we observed no significant differences between the control and anti-PD-L1 groups in plasmablast and mature B cells. Moreover, the percentage of immature B cells was low (*p* < 0.05). (**b**) In CLN, no significant differences were observed among the three groups regarding plasmablast, mature B, or immature B cells. (**c**) In SPL, the proportion of mature B cells was significantly higher in the anti-PD-L1 antibody group than it was in the control and OMP groups. OMP (+): OMP-treated group. PD-L1 (+): OMP+ anti-PD-L1 antibody-treated group. The Steel–Dwass test was used for statistical analysis. * *p* < 0.05, ** *p* < 0.01. Values are presented as mean ± SD.

**Figure 8 vaccines-13-00313-f008:**
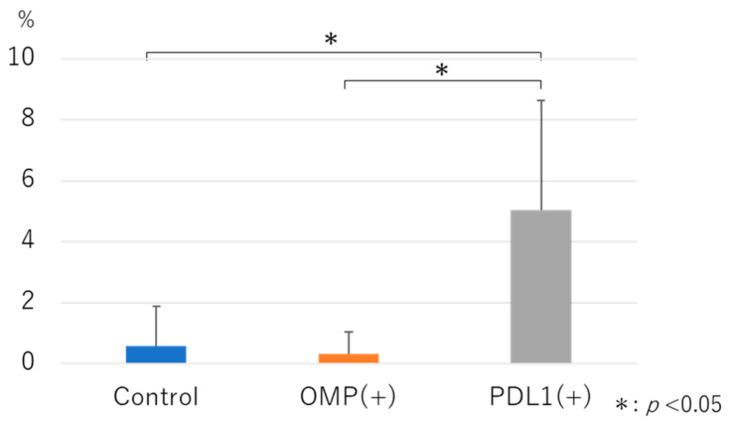
Analysis of CD3^+^CD4^+^CD185^+^ T cells after anti-PD-L1 antibody administration. The percentage of CD185+ T cells among CD3^+^CD4^+^ cells in MEM was significantly increased in the anti-PD-L1 antibody group, compared with the control and OMP groups (*p* < 0.05). OMP (+): OMP-treated group. PD-L1 (+): OMP+ anti-PD-L1 antibody-treated group. The Steel–Dwass test was used for statistical analysis. * *p* < 0.05. Values are presented as mean ± 1 SD.

**Figure 9 vaccines-13-00313-f009:**
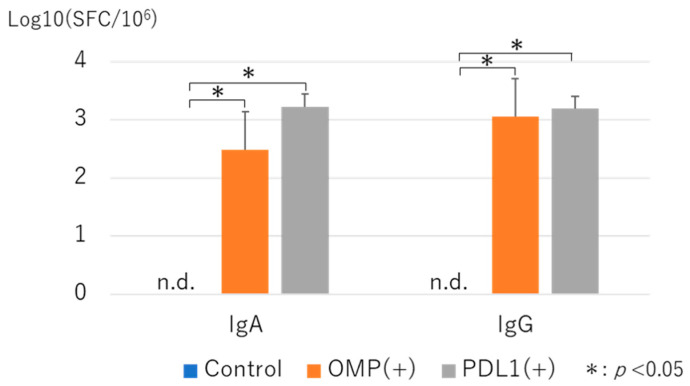
Quantification of antibody-producing cells in MEM by ELISPOT. OMP antigen-specific IgA and IgG antibody-producing cells were quantified using ELISPOT. Relative to the control group, IgA and IgG antibody-producing cells were significantly more abundant in the PD-L1 group (*p* < 0.01). Both IgA and IgG antibody-producing cells were not detected in the control group. OMP (+): OMP-treated group. PD-L1 (+): OMP+ anti-PD-L1 antibody-treated group. n.d.: not detected. The Mann–Whitney U test was used for statistical analysis. * *p* < 0.05. Values are presented as mean ± SD.

**Figure 10 vaccines-13-00313-f010:**
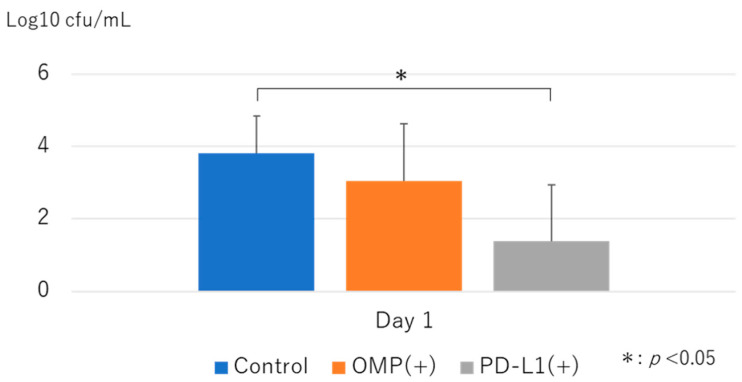
Bacterial clearance in the middle ear. Here, we quantified the number of viable nontypeable Haemophilus influenzae (NTHi) bacteria in middle ear effusions at 24 h after infection. Bacterial clearance was significantly higher in the anti-PD-L1 antibody group relative to the control group (*p* < 0.01). OMP (+): OMP-treated group. PD-L1 (+): OMP+ anti-PD-L1 antibody-treated group. The Steel–Dwass test was used for statistical analyses. * *p* < 0.05. Values are presented as mean ± SD.

**Table 1 vaccines-13-00313-t001:** Mean total number of T cells. There were no significant differences in mean total number of T cells between the control, OMP, and PD-L1 groups (i.e., MEM, CLN, SPL).

Mean Total Number of MNCs (×10^6^)	Control	OMP(+)	PD-L1
MEM	0.36 ± 0.29	0.27 ± 0.10	0.41 ± 0.20
CLN	0.80 ± 0.42	1.24 ± 0.21	1.75 ± 0.73
SPL	1.52 ± 0.82	2.21 ± 1.22	1.11 ± 0.50

## Data Availability

The original contributions presented in this study are included in the article. Further inquiries can be directed to the corresponding author.

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
