# Peer review of "Effect of Anti-Programmed Cell Death-1 Antibody on Middle Ear Mucosal Immune Response to Intranasal Administration of Haemophilus influenzae Outer Membrane Protein"

_vaccines, 2025, doi:10.3390/vaccines13030313_

Round 1
Reviewer 1 Report
Comments and Suggestions for Authors
review attached
Manuscript ID:
Effect of anti- Programmed cell death-1 antibody on middle ear 2 mucosal immune response to intranasal administration of Hae-3 mophilus influenzae outer membrane protein
General Comments
The manuscript submitted by Kazuhiro Yoshinaga et al describes the use of anti-PD-L1 Ab therapy in a vaccination study in mice regarding Haemophilus influenzae (NTHi) infection. The vaccine used in the study consists of a crude, unpurified and uncharacterised preparation from NTHi termed ‘outer membrane protein’. Anti-PD-L1 was antibody. Their key findings appear to be derived from figure that show OMP-specific IgA and IgG in the middle ear and serum respectively. The impact of the PD-L1 treatment appeared to be modest compared to the vaccine alone group. These data roughly correlated to the challenge model where there were modest drops in the bacterial load for both the vaccine alone and vaccine plus PD-L1 group. There was no statistical difference between the challenge groups.
The group used flow cytometry techniques to analyse the various T cell populations. No compelling differences were observed in the T cell populations between the groups with background (i.e. control) T cell % often similar to the treatment group.
The major conclusion that anti-PD-L1 treatment increases humoral immunity is barely supported by the data presented and there was no increased benefit to protection.
In general, I found the T cell data very difficult to interpret without the numbers of cells being included. These data should be included.
Specific comments:
Ln 31: Statements such as ‘one of the most common infectious diseases’ need to be supported by some facts – what does most common mean exactly? Better to be explicit
Ln 35-36: Perhaps some clarification is required with the link the antibiotic resistance to bacterial strains given the causative pathogen name suggest it is a virus – the subtlety might be lost on most readers.
Ln 49: the statement: Since 2017, it has been used in head and neck cancer…seems really odd and out of place - it has been used more widely than this – melanoma in particular but many other indications as well.
Ln 69: No QC provided for the vaccine. Some data should be provided that defines their batch preparation is consistent to published standard. How was the dose of OMP determined exactly?
Ln 142: For challenge study I failed to finde the administration protocol and in general I could find this for any of the studies. Administration protocols and time points for the various measurements should be added to all figures.
Throughout: The statistical test used should be included in each figure legend unless it’s the same through-out which I don’t think it can be.
In general, I found the T cell data very difficult to interpret without the numbers of cells being included. These data should be included.
How many times was the experiment repeated in figure 9?

Author Response
Comment 1:
General Comments
The manuscript submitted by Kazuhiro Yoshinaga et al describes the use of anti-PD-L1 Ab therapy in a vaccination study in mice regarding Haemophilus influenzae (NTHi) infection. The vaccine used in the study consists of a crude, unpurified and uncharacterised preparation from NTHi termed ‘outer membrane protein’. Anti-PD-L1 was antibody. Their key findings appear to be derived from figure that show OMP-specific IgA and IgG in the middle ear and serum respectively. The impact of the PD-L1 treatment appeared to be modest compared to the vaccine alone group. These data roughly correlated to the challenge model where there were modest drops in the bacterial load for both the vaccine alone and vaccine plus PD-L1 group. There was no statistical difference between the challenge groups.
The group used flow cytometry techniques to analyse the various T cell populations. No compelling differences were observed in the T cell populations between the groups with background (i.e. control) T cell % often similar to the treatment group.
The major conclusion that anti-PD-L1 treatment increases humoral immunity is barely supported by the data presented and there was no increased benefit to protection.
In general, I found the T cell data very difficult to interpret without the numbers of cells being included. These data should be included.
Response 1: Thank you for your insightful comments. In response to your concerns, we have included the total MNC counts for MEM, CLN, and SPL in Table 1, which confirm that there were no significant differences among the control, OMP(+), and PD-L1(+) groups. Based on this, we conducted statistical analyses using the proportion of specific cell populations rather than absolute cell counts. We believe that this method provides a more meaningful comparison of immune cell distributions across groups. We have clarified this point in the revised manuscript. The updated table and explanation can be found in Table 1 on page 6, line 224-226.
Specific comments:
Comment 2: Ln 30: Statements such as ‘one of the most common infectious diseases’ need to be supported by some facts – what does most common mean exactly? Better to be explicit
Response 2: Thank you for your pointing this out. The US estimates that 60% children will suffer from acute otitis media at least once by the age of 3, and up to 80% children will suffer from acute otitis media at least once in their lifetime. This suggests that AOM is one of the most common infectious diseases among children. The updated explanation can be found on page 1, line 30-32.
Comment 3: Ln 35-36: Perhaps some clarification is required with the link the antibiotic resistance to bacterial strains given the causative pathogen name suggest it is a virus – the subtlety might be lost on most readers.
Response 3: Revision:
Nontypeable Haemophilus influenzae (NTHi) is a bacterial pathogen and a major cause of AOM and respiratory infections.
Thank you for your valuable comment. We understand the confusion because of the name Haemophilus influenzae, which can be mistaken for a viral pathogen. Therefore, in the revised manuscript, we have explicitly stated that Nontypeable Haemophilus influenzae (NTHi) is a bacterial pathogen to clarify that it is not the influenza virus. The updated explanation can be found on page 1, line 34-36.
We hope this modification addresses your concern.
Comment 4: Ln 49: the statement: Since 2017, it has been used in head and neck cancer…seems really odd and out of place - it has been used more widely than this – melanoma in particular but many other indications as well.
Response 4: Thank you for your valuable feedback. In the revised manuscript, this statement has been amended to accurately reflect the broad application of PD-L1 inhibitors beyond head and neck cancer. The revised manuscript explicitly mentions their use in various advanced solid tumors, including melanoma, nonsmall cell lung cancer, renal cell carcinoma, castration-resistant prostate cancer, and colorectal cancer. In addition, I have clarified that the expansion to head and neck cancer began in 2016. I appreciate your insight and welcome any further suggestions. The updated explanation can be found on page 2, line 48-53.
Comment 5: Ln 69: No QC provided for the vaccine. Some data should be provided that defines their batch preparation is consistent to published standard. How was the dose of OMP determined exactly?
Response 5: We have revised the manuscript to clarify which QC measures were taken for the vaccine. Specifically, we performed a BCA assay to determine the protein concentration of each batch, ensuring consistency between preparations. Furthermore, we conducted the SDS-PAGE analysis to confirm the purity and integrity of the OMP. These steps were performed per previous studies from our department, and this protocol aligns with previously published standards [13,15,16].
Regarding the OMP dose determination, we based our selection (i.e., 10 μg per dose) on prior studies, including those by Kurono et al. and Hirano et al., demonstrating that optimal immune responses were observed in murine models at this concentration.
We appreciate your suggestion and hope that these revisions sufficiently address your concerns. The updated explanation can be found on page 2, line 92 through page 3, line 101.
Comment 6: Ln 142: For challenge study I failed to finde the administration protocol and in general I could find this for any of the studies. Administration protocols and time points for the various measurements should be added to all figures.
Response 6: In the revised manuscript, we have added Figure 1, illustrating the administration protocol, including time points for immunization, anti-PD-L1 antibody administration, and sample collection or bacterial challenge. This new figure therefore provides a simple, high-level overview of the experimental timeline. In addition, administration protocols and measurement time points are explicitly stated in all relevant figures. The updated explanation can be found on page 3, line 108-112.
Comment 7: Throughout: The statistical test used should be included in each figure legend unless it’s the same through-out which I don’t think it can be.
Response 7: Thank you for your valuable feedback. We have revised the manuscript to include information regarding which statistical tests were used in each figure legend. Specifically, we used the Mann–Whitney U test in Figures 2 and 9, and the Steel–Dwass test in Figures 3–8 and 10. These additions ensure clarity and transparency regarding the statistical methods employed in our analysis.
Comment 8: In general, I found the T cell data very difficult to interpret without the numbers of cells being included. These data should be included.
Response 8: Thank you for your insightful comment. Herein, we focused on presenting these data points as proportions because there were no significant differences in the total number of mononuclear cells among the three groups (i.e., Control, OMP(+), and PD-L1) in MEM, CLN, and SPL. Considerably, showing the relative percentages effectively compares immune cell distributions.
Comment 9: How many times was the experiment repeated in figure 9?
Response 9: Since we added Figure 1, Figure 9 has now been updated to Figure 10.Thank you for your inquiry regarding Figure 9. This experiment was conducted three times and included a preliminary study involving 6-month-old mice. We obtained stable and consistent results across each of these replicates. In addition, because of the high cost of reagents, with each anti-PD-L1 antibody treatment costing approximately 80,000 JPY per mouse, this experiment was performed three times to balance scientific rigor with financial feasibility. The updated table and explanation can be found in Table 1 on page 6, line 224-226.
Remark: Two-byte fonts should be changed to commonly used fonts like Times New Roman as they are not processed by most software. Thus, we have done the needful.
Reviewer 2 Report
Comments and Suggestions for Authors
The present manuscript (ID: vaccines-3470488) titled "Effect of anti- Programmed cell death-1 antibody on middle ear mucosal immune response to intranasal administration of Haemophilus influenzae outer membrane protein" provides valuable insights into how blocking PD-L1 enhances mucosal immunity in the middle ear and boosts vaccine efficacy. However, some aspects lack clarity, and the findings would benefit from simpler explanations, better figure descriptions, and a clearer connection to applications.
Abstract. The abstract is quite technical, making it difficult for non-specialists to grasp quickly. The results mention CD4+ Tfh cells and B-cell differentiation, but their broader implications aren’t clearly explained, it should be mentioned.
Introduction. The connection between PD-L1 inhibition and mucosal immunity is not well-established in the introduction. The transition from tumor immunity to mucosal vaccine responses seems to be abrupt. Add a brief transition statement explaining how PD-L1 inhibition could specifically benefit mucosal immunity, not just tumor immunity.
Methods. There is a Lack of justification for the choice of anti-PD-L1 dose and administration schedule. Was it based on previous studies? The method for B cell differentiation analysis could be explained in simpler terms for clarity.
Results. Figure 2, In case of CLN, the control vs. experimental comparisons are unclear, was there an expected increase or decrease?
Figure 5, there are no error bars or statistical significance markers in some parts of the figure.
Figure 9, the units of Y-axis (log10 CFUs/ml) should be provided on Y-axis in bigger font size.
Discussion. The authors need to clarify the broader implications of PD-L1 blockade in human vaccine development, beyond just the mouse model. Also Compare findings with existing studies on immune checkpoint inhibitors in infectious disease models to highlight novelty.
Conclusions. The authors should emphasize the translational potential of PD-L1 blockade in enhancing intranasal vaccine efficacy. Also need to include future directions, such as optimizing dosing strategies and conducting human clinical studies to validate findings.
Author Response
Comment 1: The present manuscript (ID: vaccines-3470488) titled "Effect of anti- Programmed cell death-1 antibody on middle ear mucosal immune response to intranasal administration of Haemophilus influenzae outer membrane protein" provides valuable insights into how blocking PD-L1 enhances mucosal immunity in the middle ear and boosts vaccine efficacy. However, some aspects lack clarity, and the findings would benefit from simpler explanations, better figure descriptions, and a clearer connection to applications.
Abstract. The abstract is quite technical, making it difficult for non-specialists to grasp quickly. The results mention CD4+ Tfh cells and B-cell differentiation, but their broader implications aren’t clearly explained, it should be mentioned.
Response 1: Thank you for your valuable feedback. We appreciate your suggestion to clarify the broad implications of Tfh cells and B-cell differentiation. Accordingly, we have altered the Abstract section to explicitly describe how Tfh cells support B-cell activation and antibody production, enhancing local mucosal immunity. In addition, we have highlighted the role of B-cell differentiation in increasing antibody production and improving bacterial clearance. Altogether, these modifications make the Abstract more accessible to nonspecialists while maintaining scientific accuracy. The updated explanation can be found on page 1, line 17-22.
Comment 2: Introduction. The connection between PD-L1 inhibition and mucosal immunity is not well-established in the introduction. The transition from tumor immunity to mucosal vaccine responses seems to be abrupt. Add a brief transition statement explaining how PD-L1 inhibition could specifically benefit mucosal immunity, not just tumor immunity.
Response 2: Thank you for your insight. We appreciate your suggestion to establish a clear connection between PD-L1 inhibition and mucosal immunity. Accordingly, the Introduction section has been revised to include a transition statement that highlights the potential role of PD-1/PD-L1 blockade in enhancing vaccine response. The updated explanation can be found on page 2, line 58-70.
Comment 3: Methods. There is a Lack of justification for the choice of anti-PD-L1 dose and administration schedule. Was it based on previous studies? The method for B cell differentiation analysis could be explained in simpler terms for clarity.
Response 3: The dose and administration schedule of anti-PD-L1 antibody used herein were based on a previously published protocol by Channappanavar et al. [18], who reported that 300 μg of anti-PD-L1 antibody was administered intraperitoneally on days 1 and 3 (i.e., before and after viral infection). Herein, we used a dose of 250 μg per administration following preliminary experiments, which elicited a sufficient immune response. In addition, given the high cost of the reagent (80,000 JPY per mouse), we selected this dose to optimize efficacy and resource efficiency.
Comment 4: Results. Figure 2, In case of CLN, the control vs. experimental comparisons are unclear, was there an expected increase or decrease?
Response 4: Thank you for your comment. Since we added Figure 1, Figure 2 has now been updated to Figure 3. Regarding the CLN data, we observed considerable variability in the control group. Although the mean value appeared high, we found no statistically significant difference between the means of the OMP and PD-L1 groups. We have clarified this in the revised manuscript to improve the interpretation of the data. The updated explanation can be found on page 6, line 230-234.
Comment 5: Figure 5, there are no error bars or statistical significance markers in some parts of the figure.
Response 5: Thank you for your careful review. Since we added Figure 1, Figure 5 has now been updated to Figure 6. We acknowledge the inconsistency in our description of statistical significance in the text. The previous statement regarding CD19+ cells in CLN (i.e., 32.2 ± 4.5 vs. 57.4 ± 9.3, 32.2 ± 4.5 vs. 48.46 ± 8.8; p < 0.05) was erroneous, as we found no significant difference between the control and anti-PD-L1 antibody groups in CLN. We have rectified this error in the revised manuscript. The updated explanation can be found on page 9, line 295-296.
Comment 6: Figure 9, the units of Y-axis (log10 CFUs/ml) should be provided on Y-axis in bigger font size.
Response 6: Figure 9: Since we added Figure 1, Figure 9 has now been updated to Figure 10. We have increased the Y-axis font size in Figure 10 to improve readability and ensure the units (log10 CFUs/ml) are clearly visible. The updated Figure can be found in Figure10 on page 11.
Comment 7: Discussion. The authors need to clarify the broader implications of PD-L1 blockade in human vaccine development, beyond just the mouse model. Also Compare findings with existing studies on immune checkpoint inhibitors in infectious disease models to highlight novelty.
Response 7: Thank you for your feedback. In response, we have revised the Discussion section to clarify the broad implications of PD-L1 blockade for human vaccine development (i.e., beyond the mouse model). The updated explanation can be found on page 11, line 374 through page 12, line 389.
We have emphasized the translational potential of the PD-L1 blockade by discussing its potential role in enhancing mucosal vaccine efficacy in humans, particularly via improved humoral immune responses.
We have compared our findings with existing studies on immune checkpoint inhibitors in infectious disease models. These studies include Velu et al., who developed an SIV model, and Läubli et al., who developed an influenza vaccine study in patients with cancer.
We have refined the discussion on PD-1/PD-L1 blockade in vaccine responses by distinguishing between its effects during chronic infection and its relatively unexplored role in primary vaccine-induced immunity.
Overall, these revisions connect our findings and the results of other contemporary research studies, thereby highlighting the novelty and clinical relevance of our study. We appreciate your valuable comments, which have helped improve the clarity and impact of our manuscript.
Comment 8: Conclusions. The authors should emphasize the translational potential of PD-L1 blockade in enhancing intranasal vaccine efficacy. Also need to include future directions, such as optimizing dosing strategies and conducting human clinical studies to validate findings.
Response 8: We have revised our conclusions to emphasize the translational potential of PD-L1 blockade for mucosal vaccine development and have noted important future research directions, including dose optimization and the need for clinical studies. The updated explanation can be found on page 13, line 466-470.
Reviewer 3 Report
Comments and Suggestions for Authors
The main causative agents of acute otitis media are Streptococcus pneumoniae and Haemophilus influenzae, as well as Moraxella catarrhalis and Streptococcus pyogenes. At the same time, it is necessary to note a higher frequency of H. influenzae in comparison with acute bacterial sinusitis. Otitis media caused by H. influenzae is often characterized by subclinical course with the absence of pronounced clinical symptoms and at the same time with a pronounced effect on the physiology and morphology of the mesenteric epithelium due to adhesins and endotoxin of H. influenzae. There is currently a high demand for non-typable H. influenzae (NTHi ) vaccines. There are several options for the composition of such vaccines. Vaccine composition decisions must take into account the antigenic variability of NTHi, so that even complex immunogens such as whole bacteria would preferably have a tailored antigenic composition. Here, the authors used the mouse NTHi vaccination model to investigate the effect of anti-PD-L1 antibodies on the level of local and systemic humoral response to the vaccine. The work has been thorough and in compliance with all the necessary validation criteria. The conclusions drawn by the authors of the study are fully justified by the results obtained.
Author Response
Comments: The main causative agents of acute otitis media are Streptococcus pneumoniae and Haemophilus influenzae, as well as Moraxella catarrhalis and Streptococcus pyogenes. At the same time, it is necessary to note a higher frequency of H. influenzae in comparison with acute bacterial sinusitis. Otitis media caused by H. influenzae is often characterized by subclinical course with the absence of pronounced clinical symptoms and at the same time with a pronounced effect on the physiology and morphology of the mesenteric epithelium due to adhesins and endotoxin of H. influenzae. There is currently a high demand for non-typable H. influenzae (NTHi ) vaccines. There are several options for the composition of such vaccines. Vaccine composition decisions must take into account the antigenic variability of NTHi, so that even complex immunogens such as whole bacteria would preferably have a tailored antigenic composition. Here, the authors used the mouse NTHi vaccination model to investigate the effect of anti-PD-L1 antibodies on the level of local and systemic humoral response to the vaccine. The work has been thorough and in compliance with all the necessary validation criteria. The conclusions drawn by the authors of the study are fully justified by the results obtained.
Response:
We sincerely appreciate your thoughtful review and positive feedback on our study. We are particularly grateful for your recognition of the importance of developing an effective non-typeable Haemophilus influenzae (NTHi) vaccine and the challenges posed by its antigenic variability.
As you highlighted, our study aimed to explore the role of PD-L1 blockade in enhancing both local and systemic humoral responses following intranasal NTHi vaccination. Our findings suggest that anti-PD-L1 therapy could be a promising strategy to improve mucosal vaccine efficacy, which may have broader implications for vaccine development against respiratory pathogens.
Once again, we appreciate your supportive comments and valuable insights.
Round 2
Reviewer 1 Report
Comments and Suggestions for Authors
Manuscript ID:
Effect of anti- Programmed cell death-1 antibody on middle ear 2 mucosal immune response to intranasal administration of Hae-3 mophilus influenzae outer membrane protein
General Comments
The manuscript submitted by Kazuhiro Yoshinaga et al describes the use of anti-PD-L1 Ab therapy in a vaccination study in mice regarding Haemophilus influenzae (NTHi) infection. In the revision the authors have adequately addressed specific comments raised by the reviewer.
The intriguing study shows that the addition of anti-PD1 therapy may be beneficial when administered alongside vaccination so may be of some interest to researchers and developers in the specific field.
The data presented was not supported by clear mechanistic data, comparisons with other checkpoint blockage inhibitors or the other antigens which would make the investigation generally relevant to more researchers ab developers.
Author Response
General Comments
The manuscript submitted by Kazuhiro Yoshinaga et al describes the use of anti-PD-L1 Ab therapy in a vaccination study in mice regarding Haemophilus influenzae (NTHi) infection. In the revision the authors have adequately addressed specific comments raised by the reviewer.
The intriguing study shows that the addition of anti-PD1 therapy may be beneficial when administered alongside vaccination so may be of some interest to researchers and developers in the specific field.
The data presented was not supported by clear mechanistic data, comparisons with other checkpoint blockage inhibitors or the other antigens which would make the investigation generally relevant to more researchers ab developers.
Response: Thank you very much for your thoughtful and constructive comments on our manuscript. We appreciate your recognition of the value of our study and the potential benefits of combining anti-PD-L1 therapy with vaccination in the context of Haemophilus influenzae (NTHi) infection.
We also acknowledge your point regarding the lack of clear mechanistic data, as well as the absence of comparisons with other checkpoint inhibitors and antigens. We agree that these aspects would enhance the relevance and applicability of our findings. We plan to address these points in future studies, where we aim to investigate the underlying mechanisms more thoroughly and compare our results with other checkpoint inhibitors and antigens to broaden the scope of the study.
Thank you again for your valuable feedback. We look forward to further improving our work based on your suggestions.
Best regards,
Kazuhiro Yoshinaga et al.
Reviewer 2 Report
Comments and Suggestions for Authors
The authors have satisfactorily responded to my comments in the revised manuscript.
Comments on the Quality of English LanguageThe authors have adequately addressed my comments. Please proceed with your final decision.
Author Response
The authors have satisfactorily responded to my comments in the revised manuscript.
Response: Thank you for your positive feedback. We are pleased to hear that our responses to your comments have satisfactorily addressed your concerns in the revised manuscript. Your input has been invaluable in improving the quality of our work.
We appreciate your time and effort in reviewing our manuscript and look forward to any further suggestions you may have.
Best regards,
Kazuhiro Yoshinaga et al.